# Barriers in the Management of Obesity in Mexican Children and Adolescents through the COVID-19 Lockdown—Lessons Learned and Perspectives for the Future

**DOI:** 10.3390/nu15194238

**Published:** 2023-09-30

**Authors:** Paulina Arellano-Alvarez, Brenda Muñoz-Guerrero, Alejandra Ruiz-Barranco, Nayely Garibay-Nieto, Ana María Hernandez-Lopez, Karina Aguilar-Cuarto, Karen Pedraza-Escudero, Zendy Fuentes-Corona, Erendira Villanueva-Ortega

**Affiliations:** 1Pediatric Obesity Clinic at Child Welfare Unit (UBI, Acronym in Spanish) of The Hospital General de México “Dr. Eduardo Liceaga”, Mexico City 06720, Mexico; dra.paulina.arellano@gmail.com (P.A.-A.); brendmg29@gmail.com (B.M.-G.); aleruba30@hotmail.com (A.R.-B.); gngaribay@hotmail.com (N.G.-N.); anahello7@gmail.com (A.M.H.-L.); kari.cuarto@gmail.com (K.A.-C.); karenpedrazaescudero@outlook.es (K.P.-E.); zendy_ff@outlook.es (Z.F.-C.); 2Master’s and Doctorate Program in Medical, Dental and Health Sciences, Universidad Nacional Autónoma de México, Mexico City 04510, Mexico

**Keywords:** childhood obesity, COVID-19, lifestyle changes, lockdown, adolescents

## Abstract

The COVID-19 lockdown caused health system issues, including the need for long-term care for patients with conditions like childhood obesity. We wanted to know how the lockdown had changed our patients’ health and which variables had greater influence in preventing and managing overweight and obesity in kids and teens during and after the lockdown. Methods: Our study comprised two phases. The initial descriptive cross-sectional phase included surveys of children who are overweight or obese and their parents/guardians from the Pediatric Obesity Clinic at the Child Welfare Unit (UBI, acronym in Spanish) in the Hospital General de México “Dr. Eduardo Liceaga” (n = 129). The participants were studied to explore changes in lifestyle, physical activity, sleep patterns, eating behaviors, food consumption, anxiety, and depression. In the second phase, the biochemical, body composition, and anthropometric parameters of 29 pre-COVID-19 patients were compared before and after the lockdown. Results: The survey showed more moderate anxiety and depression, alterations in sleep, physical inactivity, and an increase in the consumption of animal products, fruits, cereals, tubers, sugary drinks, and ultra-processed food. In the study’s comparative phase, we observed a substantial increase in BMI z-score (*p* = 0.01), waist circumference (*p* < 0.001), fat mass (*p* < 0.001), percentage of adiposity (*p* = 0.002), and basal glucose (*p* = 0.047) and a drop in lean mass (*p* = 0.008). Conclusions: The pandemic led to a loss of routines and socioeconomic changes that made it difficult to address weight and obesity in young people. The results show that managing obesity in our patients involves considering both lifestyle and the social environment. This encourages us to consider a comprehensive and personalized approach.

## 1. Introduction

COVID-19 (SARS-CoV-2) caused a pandemic in 2019 [1], affecting health through both the virus itself and the social distancing measures put in place to stop its spread. This led to non-communicable diseases that might not yet be fully identified [2].

The effects of COVID-19 and social distancing have affected the health, social, economic, and mental health fields [3,4]. We are living in a “new normality” that is not comparable with a similar precedent. The digital age has allowed us to connect more easily, but it has also increased stress and anxiety because of teleworking overload and inaccurate information [5,6,7,8].

Social distancing had a negative effect on obesity and other chronic diseases because medical care was devoted to health emergencies, leaving these patients without follow-up [9]. This type of patient suffered double damage because, as we now know, the SARS-CoV-2 infection generated multi-organ damage, particularly in patients previously affected by chronic conditions, such as patients with obesity and its comorbidities (type 2 diabetes, arterial hypertension, cancer, cardiovascular diseases, hepatic steatosis, chronic bronchitis, and others) [10].

The child population was no exception. Given that individual, interpersonal, and community environment factors together influence the incidence and prevalence of obesity, the pandemic worsened the weight and obesity crisis because of routine disruptions, the limited availability of public spaces for physical activity, and the reduced active play, which cut off a healthy lifestyle [11,12,13,14].

The increase in obesity rates during the pandemic was unprecedented. In May 2020, Brown University conducted a study that used a simulation model to predict an increase in childhood obesity rates and BMI z-scores because of COVID-19 confinement; this study was based on previous research that showed an increase in these two parameters during summer vacation in children who are overweight and obese [14,15,16,17]. The study looked at four possible outcomes, each increasing COVID-19 confinement by 2 months. The results showed that obesity prevalence could rise from 0.64 to 2.37 percentage points while the BMI z-score could rise from 0.053 to 0.718 points [14]. In Mexico, the social distancing measures for the child population lasted approximately 15 months; in some cases, by decision of the parents, the time could be even longer [18]. We do not know the effect of confinement on obesity after COVID-19, as no longitudinal studies have yet been published.

To reduce factors associated with obesity, it is critical to intervene in routine acquisition, which includes establishing eating schedules, exercising regularly, reducing sedentary activities, and setting sleep routines [19,20]. Mexico has been heavily affected by obesity and the pandemic and lockdown contributed to the disrupted routines that are the gold standard in obesity management, increasing the challenge; therefore, we must assess its effects on children and adolescents who are overweight and obese [21,22,23,24].

The aim of the study was to identify the environmental, social, and psychological factors that had the greatest impact on pediatric patients who are overweight and obese during the COVID-19 lockdown. In addition, we attempt to document if the loss of lifestyle during the lockdown had an adverse effect on the anthropometric parameters, metabolic markers, and body composition changes of children and adolescents who are overweight and obese. This analysis will help us evaluate the changes in the management of childhood obesity in our population and the difficulties that healthcare professionals must deal with in treating these patients [24].

## 2. Materials and Methods

This study comprised 2 phases. The first phase had a cross-sectional and descriptive design, involving surveys of children and adolescents who are overweight and obese, and their guardians. In a second phase, we considered a comparative longitudinal design for a subgroup of patients previously known in our unit before the lockdown due to COVID-19; for this subgroup, we had anthropometric and biochemical markers before and after confinement, and we could analyze the changes because of this atypical event.

We conducted this study in the Pediatric Obesity Clinic at the Child Welfare Unit (UBI, acronym in Spanish) Hospital General of México Dr. Eduardo Liceaga from November 2021 to April 2022. This clinic is a specialized center in holistic medical care for children and adolescents who are overweight or obese. Because our hospital is one of the biggest in Mexico City, during the confinement it focused only on COVID-19 patients, and the reopening to other medical care activities was gradual; therefore, the recruitment for the study was slow, and the sample size was limited. We invited all patients between 8 and 17 years old who came to our clinic with consecutive diagnoses of being overweight and obese, excluding patients with syndromic obesity or secondary to endocrinopathies and with incomplete anthropometric variables.

We included 129 patients who are overweight and obese, according to the Centers for Disease Control and Prevention (CDC) guidelines. We defined a person as being overweight when they have a body mass index (BMI) ≥ 85th percentile and < 95th percentile, and obesity as a BMI ≥ 95th percentile. We also surveyed 119 guardians (the same family member accompanied 20 patients). The ethics and clinical research committees of the Hospital General de México Dr. Eduardo Liceaga approved the study with the registration number DI/21/303/03/64 (date of approval 10 November 2021).

We invited the guardians and patients to take part in this study. After they accepted the assent and informed agreement, they completed a questionnaire about the effects of the COVID-19 lockdown on their lifestyles and their social and mental health environment.

The survey included six questionnaires. Five of them were for patients and one was for their relatives. Among the five questionnaires for patients were EAT, BIRLESON, and SCARED, which assessed eating disorders, depression, and anxiety, respectively. This last topic explored mental health after the confinement; the patients reported their symptoms and experiences during pre- and post-confinement.

Another survey for patients had five categories: lifestyle, diet, sedentary behavior, physical activity, and sleep patterns, as well as an online medical care evaluation. With the guidance of a nutritionist, they finished the last questionnaire, which assessed their feeding frequency and their preferences for food groups. In general, the questionnaires sought to investigate confinement experience. The guardians answered questions about lifestyle changes, economic and job losses, family life during the lockdown, the loss of health, eating habits, and virtual healthcare. Fill time for all the surveys was approximately 20 to 30 min for both patients and guardians.

Of the total sample, 29 children and adolescents had been our patients before the lockdown. Therefore, we had anthropometric, metabolic, and body composition data enabling us to evaluate these same variables post-lockdown for comparison. Clinical data collected before and after confinement were age, sex, weight, height, BMI, waist circumference, and blood pressure. Routinely, a mechanical scale and stadiometer SECA model 2841300379 were used to assess weight and height. We measured the waist circumference with a precision of 0.1 mm using a 2 m Lufkin flexometer at the midpoint of the distance between the last costal cartilage and the anterior superior iliac crest. Blood pressure was assessed in the right brachial artery using a Welch Allyn model DS44-MC manual sphygmomanometer with a baumanometer, adjusted to the size of the subject’s arm, with prior 5 min of rest. The BMI was calculated according to the Quetelet equation.

We evaluated metabolic variables with a 12 h fast. Commercial kits were used to measure glucose, alanine aminotransferase (ALT), aspartate aminotransferase (AST), total cholesterol, high-density lipoprotein (HDL), low-density lipoprotein (LDL), and triglycerides, using enzymatic methods. Insulin was measured by chemiluminescence analysis.

We evaluated body composition with a Jawon ioi 353 bioimpedance meter with 5 electrodes, with frequency ranges of 5, 50, and 250 kHz validated with the isotope dilution technique and Dual Energy X-ray Absorptiometry (DEXA).

### Statistical Analysis

We expressed the quantitative variables as means and standard deviations, while the categorical variables were expressed as frequencies and percentages.

To evaluate differences in anthropometric, body composition, as well as metabolic results, before and after the lockdown, we conducted a paired T-test.

Statistical analysis was performed using IBM SPSS Statistics version 28.0 for Mac software (IBM, Armonk, NY, USA). We considered statistical significance if the *p*-value was less than 0.05.

## 3. Results

### 3.1. Demographics

Our study included 129 children and adolescents from 8 to 18 years of age (mean = 13, SD = 3); 52% were male. A total of 75% had a diagnosis of obesity and 25% were patients who were overweight according to the diagnostic criteria of the CDC.

We also considered guardians of the patients (n = 119), as 20 patients were siblings and had the same guardian. The age of the parents had a mean = 40 years and an SD = 8; 64% of the parents were in the age range of 36 to 60 years, and the most frequent companion was the mother (87%). Guardians had a middle school in 38% of cases and high school in 31% of cases; in terms of marital status, the parents were married in 41% of cases, the frequency of single-parent families reached up to 31% (17% single and 9% separated), and families with over 10 members were uncommon, with only 5% falling into this category. Most of the families had two to four members (57%) (Table 1).

### 3.2. Lifestyles

We studied how social distancing measures affected the lifestyles of pediatric patients who are overweight and obese during the COVID-19 lockdown.

We found that 98% of our patients believed that COVID-19 could be a serious disease, but, interestingly, only 76% changed their lifestyles due to it (Figure 1A,B).

Our patients were asked to consider the factors contributing to changes in their lifestyles; routine changes were indicated by 29%, with 21% attributing it only to diet (schedules and frequencies), 15% attributing it only to improving their health, 10% attributing it to emotional aspects, and 10% attributing it to physical activity decrease (Figure 1C). Up to 89% of patients reported having increased their weight to a greater or lesser extent (Figure 1D).

When we questioned the guardians, they also reported lifestyle changes in 80% of cases. In up to 46% of cases, habits worsened. It is important to highlight the fact that, during confinement, 43% experienced an increase in work time greater than 8 h per day (Figure 1E–G). The lifestyle changes that family members reported were described by the guardians as relating mainly to major changes in daily routine (28%), finances (job and economic losses) (24%), and health effects (20%). (Up to 11% experienced the loss of a family member due to COVID-19.)

With regard to food security, up to 39% of children lost food support usually provided by schools and 77% of family members reported a loss of income (Figure 1H,I).

### 3.3. Food and Feeding

We researched changes in eating habits and food group consumption during the lockdown, considering restrictions and family context.

Up to 53% of the patients acknowledged that their eating habits had worsened during confinement (Figure 2A); 48% expressed a constant feeling of hunger (Figure 2B), although we found that there was an omission of meals or snacks in 36% of patients, while another 30% had at least one additional meal or snack time (Figure 2C). Regarding behavior patterns, 31% reported feeling hungry among the principal foods (Figure 2D). Fried food and carbohydrates were the most preferred food and junk food (Figure 2E), and 73% of the patients reported water consumption of less than 2 L per day (Figure 2F).

Regarding feeding behavior, 44% of children and adolescents reported that they consumed food while in front of screens (browsing the Internet or watching television). During the confinement, 56% of the parents encouraged restrictive behaviors in their children who are overweight or obese, while only 4% favored healthy eating behaviors. With respect to food supply, 41% went to local markets, 23% went to street markets close to their homes, 27% went to nearby grocery stores, and only 9% went to supermarkets.

In terms of the preference for food groups considering their previous intake, patients reported consumption of 57% more animal products, 63% more fruits, and 61% more cereals and tubers during the confinement. Although up to 53% of patients increased their consumption of vegetables, another 31% decreased their intake (Figure 3).

With regard to sweetened beverages (soda-type), the patients reported an increase in 45% of cases, while 35% maintained the same pattern of consumption. During the lockdown, more people cooked at home (54% increase), but, paradoxically, they consumed more unhealthy ultra-processed foods (51% increase). The number of people who ate outside of the home before and during the confinement was the same (78% of patients) (Figure 3).

### 3.4. Sleep Patterns, Sedentary Lifestyle, and Physical Activity

Many patients experienced alterations in their pattern and sleep routines. Up to 45% of patients slept more during confinement, while 27% had trouble falling asleep (27%) and 9% took daytime naps (Figure 4C). The most remarkable change was in the jet lag for time to go to sleep, which before confinement was mostly before 22:00 h in 67% of cases and between 22:00 h and 00:00 h in 24% of children and adolescents. During the lockdown, the most frequent bedtime was between 22:00 h and 00:00 h in 58% of cases, followed by the 00:00 h to 03:00 h schedule of 33% of those surveyed. This jet lag also changed the wake-up time, which before the confinement was mostly before 08:00 h (76%). Meanwhile, during the lockdown, 40% of patients woke up before 08:00 h, 35% woke up from 8:00 h to 10:00 h, and 21% woke up from 10:00 h to 12:00 h (Figure 4A,B).

Regarding screen time, 54% of patients used devices for over 2 h for school activities, while 46% of them did the same for recreational purposes (Figure 4E,F). The most frequent entertainment activities were watching videos, series, or movies (26%), followed by online games (19%), and listening to music while sitting down (15%) (Figure 4D). The most used device was the cell phone at 64%, followed by the television at 15% and the personal computer at 7%.

In terms of physical activity, before the lockdown, 41% of our patients went to the gym, 11% ran, and 11% swam. During the lockdown, up to 69% did not do any activity, 11% went to the gym, and 8% ran (Figure 4H,I). When we asked the patients about what could have helped them maintain physical activities during the lockdown period, they reported that having physical space (29%), being thin (18%), and having company and motivation (16%) were the main factors that would have helped. Many patients struggled to exercise during the lockdown due to a lack of motivation (47%), lack of time (18%), and lack of space (14%). A worrying situation was that bullying associated with weight was also carried out by family members during the lockdown (Figure 4I,J).

### 3.5. Mental Health

Our mental health research focused on the occurrence of depression and anxiety. Depressive symptoms before and after confinement were the same (40%). However, during confinement, 8% of our patients showed depression that was not previously present (Figure 5A).

Before the lockdown, 47% of our patients experienced anxiety. During the lockdown, moderate anxiety increased from 32% to 41%. Similar to our observances about depression, there was a predominance in the female sex, without reaching statistical differences (Figure 5B).

### 3.6. Changes in Body Composition, Metabolic Markers, and Anthropometric Measurements

We surveyed 29 children and adolescents who were patients in the Pediatric Obesity Clinic before the lockdown to compare their clinical, body composition, and metabolic parameters before and after social distancing.

We found that the BMI z-score (*p* = 0.01), waist circumference (*p* < 0.001), fat mass percentage (*p* < 0.001), adiposity index (*p* = 0.002), and fasting glucose (*p* = 0.047) increased significantly, while the lean mass percentage declined (*p* = 0.008) (Table 2).

### 3.7. e-Health

Due to COVID-19, we had to implement a strategy of remote medical care using telemedicine. When we asked our patients about this service, over 50% said that they liked it because it helped them save time and money. However, 52% of parents and 63% of patients did not like it because they thought it felt impersonal and incomplete due to a lack of physical examination.

### 3.8. General Perception of the Pandemic and Lockdown

We asked our patients and their parents to write about their pandemic and confinement experiences. The answers showed an increased understanding of the importance of health, hygiene, social distancing, and the care of family members. Some also highlighted the value of family and loved ones.

## 4. Discussion

This study aimed to identify the environmental, social, and psychological factors linked to obesity that experienced the most changes during the COVID-19 lockdown. As we have already shown, patients who are overweight and obese who visited our Pediatric Obesity Clinic experienced an increase in anthropometric measurements, metabolic parameter alterations, and body composition changes because of the COVID-19 lockdown in our country.

Poor eating habits and less physical activity affected our patients’ health. Many blamed the changes on disrupted routines in school and work. A change in sleep patterns – notably, a shift in bedtime to a later hour – was a key factor for 96% of patients, with a three-fold increase in frequency. Another factor that added to the sedentary lifestyle was an increase in screen hours related to school activities.

Up to 76% of our patients reported having changed their lifestyles during the pandemic. However, 46% experienced negative changes. In this sense, Di Renzo in 2020 [25] interviewed an Italian population, among which 46% perceived changed habits, mostly worse (37%). Social distancing negatively affected the economy, causing 97% of patients to lose their jobs or self-employment. Interestingly, over 30% of these patients worked to improve their health by changing their lifestyle.

In our study, we noticed an increase in the number of patients who ate from food groups like fruits, protein, cereals, and tubers during the lockdown. This differs from the Di Renzo study [25] and is an interesting finding given that his population predominantly follows the Mediterranean-type diet. This leads us to think that lockdown-related changes in eating patterns were influenced mostly by typical diets and the idea of improving one’s diet or eating from a healthy menu to prevent the spread of COVID-19. In our case, the Mexican diet is usually high in cereals and tubers, which led our patients to increase their consumption during the lockdown.

Similar to our study, Ruiz Rosso et al. 2020 [26] found that people in Europe and Latin America ate more fruits and some vegetables. Regardless of the high consumption of other not-so-healthy food groups in the confinement, we believe that the explanation for this behavior is that these nutrients do not require preparation and are ready and easy to eat. Perhaps a few people consumed them for health reasons, desiring to boost their immune system against COVID-19, including nutraceuticals like zinc, vitamin D, and quercetin, among others [27,28]. Paradoxically, during the lockdown, people consumed more ultra-processed products and sweetened beverages. We believe that the use of ultra-processed products was due to panic buying, the fear of consuming food prepared by people who might generate contagion, or even the period of confinement, at whose end we perhaps relaxed our healthcare measures.

During our investigation of factors related to food insecurity, we found that 70% of parents of our patients reported a decrease in their income, and up to 39% of children and teenagers no longer received food support from schools during the lockdown. We did not formally evaluate the socioeconomic conditions of our patients in this study, though we believed that the behavior of our patients could be like that in Adams et al.’s report; they found that families with food insecurity in the US consume more ultra-processed products and that severe food insecurity, caused that parents of patients who are overweight, causes restrictive behaviors [29].

With respect to eating behaviors, we found that up to one-third of our patients increased their number of meals per day, snacks, or main meals during the lockdown. This finding aligns with that of Di Renzo and colleagues in 2020 [25], as they reported a 44% increase in food intake in the general population. We believe that the increase in meals or snacks could be attributed to behavioral changes during the confinement, such as boredom or anxiety [30].

In our study the patients’ inactivity increased by 26% during the lockdown, reaching a frequency of 69% of children and adolescents who remained physically inactive. This frequency was like that reported by Ruiz Rosso et al. 2020 [31], as they found 79.5% inactivity among subjects in a multinational study of adolescents. Interestingly, this frequency was higher in Latin American countries, where the numbers reached 82.7%. In our study, patients faced limitations in their ability to exercise during confinement due to a lack of space, their physical condition, or a lack of company. This factor highlights a difficulty that many children and adolescents with obesity face, even in other conditions before the pandemic, as made clear by Tan and collaborators in a Chinese population; they showed that access to public spaces in the urban environment influences the individual health of these patients [13].

One routine that was most affected during confinement was sleeping habits. The patients reported that, on average, they went to bed 2 h later than they did before the confinement. Most of the patients went to bed between 22:00 h and midnight. Our observations match those of Cellini et al.’s report in 2021 at Italian schools; they observed a 2-h delay in bedtime that did not affect sleep duration [32]. The loss of school routines and the increased use of screen time might explain this change in sleeping habits. We do not currently know if the change in our patients’ sleep schedules during confinement will affect their health status in the medium or long term, but previous research evidence shows that postponing sleep (social jet lag) is associated with poor diet choices and several cardiometabolic health problems [33]. The Bingqian Zhu et al. meta-analysis [34] shows that delaying bedtime by 2 or more hours increases the risk of being overweight, is associated with larger waist circumference, and impaired glucose levels at 2 h in the challenge test.

Unlike the reports of other research groups with respect to sleep disorders during the lockdown, ours did not find a statistical difference in the number of hours (7–9 h) pre- or during confinement in our children and adolescents; in adults, some studies have reported disturbances in sleep quality linked predominantly to mental health disorders. Other variables linked to sleep quality disturbances have been associated with an increase in protein consumption and a decrease in vegetable consumption; in our study, the consumption of cereals, fruits, protein, and vegetables was almost equivalent, and we did not find associations with social jet lag [35,36].

Regarding mental health, the frequency of depression and depression symptoms increased in 48% of the patients, and anxiety in 54%; we even observed a greater trend in females. We cannot find statistical significance; other studies have studied depression with obesity, but the evidence is controversial. As in Quek et al., we found that female adolescents are more likely to have depression, but factors beyond the presence of obesity could influence the depression risk [37,38,39].

We intentionally sought data related to depression and anxiety. Anxiety was the most common condition, with a frequency of 54%. This is higher than that reported by a study from the Netherlands, which found a frequency of 32% among children with obesity [40]. Our higher numbers might be attributed to the moment when the surveys were administered; the Netherlands study was at the beginning of the lockdown, while our study was at the end. Another hypothesis involves quality of life, given that our population mostly has lower economic incomes and fewer resources to stay at home. On this issue, our major limitation is recalling bias, as we did not have mental health statistics for all the patients approached in the study before the pandemic, and the pre-lockdown information was collected retrospectively.

Our clinic used telemedicine mostly for psychological attention during the confinement. In our survey, up to 63% of our patients reported that this type of care was impersonal and there was a significant loss of follow-up. A study by Linardon et al. 2020 reported that though online care is less expensive and more convenient, up to 70% of patients with eating disorders preferred face-to-face care. This combination of advantages and disadvantages in telemedicine might improve patients’ perception of e-health in hybrid models of care [41].

The major limitations of this study were the sample size, the recall bias about habits before the lockdown, and the mental health condition before confinement. Our strengths were the face-to-face survey and the objective comparison of clinic, metabolic, and anthropometric variables in a patient subgroup.

Our study found that COVID-19 confinement contributed to an increase in the occurrence of excess weight and obesity in children and adolescents due to negative effects on eating behavior, physical activity, and sleep hygiene, as well as an increase in sedentary activities, which, all together, led to clinical effects on adiposity parameters and glucose levels. Without also forgetting the effect on mental health.

Our population belongs to one of the most unprotected sectors nationwide. Although they have universal health insurance, access to these services is affected by both geographic issues and online medical care services. This was probably a contributor to obesity progress, as has been described among some populations and ethnic groups by Traore et al. [42].

Obesity is a multifaceted health issue. These findings suggest that there is no one-size-fits-all solution and that several components could affect this condition, which traditionally has been attributed only to physical activity and eating behaviors. Now we know that the situation is more complex and that all the activities or behaviors of patients are interconnected and influence their health; therefore, in theory, individual intervention with a conductive behavioral approach could affect the rest of them [43].

We need a comprehensive approach to treating childhood obesity, one that considers both individual and broader factors. The most visible approach is to focus on the patient’s needs, including diagnosis, mental health, and strengths or weaknesses. The COVID-19 confinement made us realize that factors beyond our control affect healthy lifestyle habits. Health policies should focus on changing personal, family, and social environments.

COVID-19 showed us the need to be ready to adapt to new challenges and solve problems quickly and in real time. Likely, we are still not aware of everything that this event has changed in the different aspects of our lives. This study caused us to reflect on the notion that treating obesity before and after COVID-19 requires empathy, compassion, and no stigmatization of patients, allowing us to provide personalized and quality care for patients and their families while we are still living in and adapting to this new reality.

## Figures and Tables

**Figure 1 nutrients-15-04238-f001:**
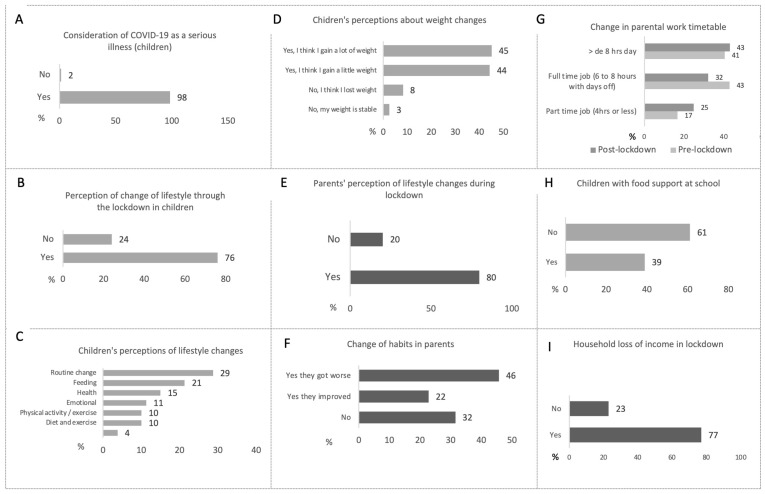
Frequencies of results on the items referring to changes in the lifestyles of children-adolescents who are obese and overweight and their family members surveyed in the Child Well-being Unit. (**A**) Proportion of children who consider COVID-19 to be a significant illness. (**B**) Percentage of children who think they changed their lifestyles during the lockdown. (**C**) Children’s perspectives on the changes in their lifestyles. (**D**) Patients’ perception of weight changes. (**E**) Parental perspective on lifestyle modifications during the lockdown. (**F**) Parental perspective regarding their own habit progress. (**G**) Change in parental work timetable due to the lockdown. (**H**) Proportion of children benefiting from school food support. (**I**) Percentage of households experiencing a decline in economic income during the lockdown. Results expressed in percentages, *n* = 129.

**Figure 2 nutrients-15-04238-f002:**
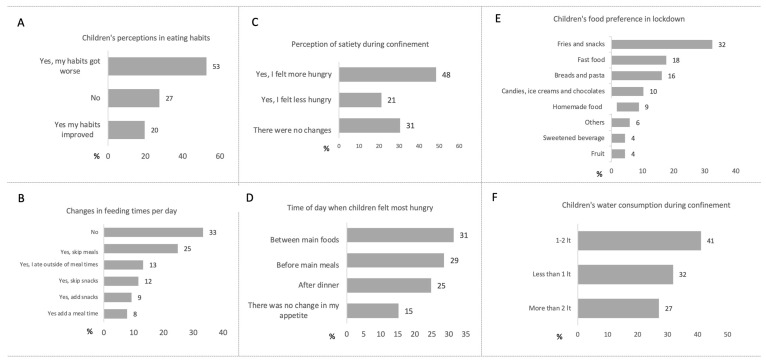
Frequencies of results on the items referring to changes in the diet of children−adolescents who are obese and overweight and their relatives surveyed at the Pediatric Obesity Clinic in the Child Well-being Unit. (**A**) Children’s perception of their eating habits. (**B**) Change in feeding times per day. (**C**) Modification in the perception of satiety in children. (**D**) Time of the day when children reported the most significant appetite. (**E**) Children’s food preference during confinement. (**F**) Water consumption reported by patients during confinement. Results expressed in percentages, *n* = 129.

**Figure 3 nutrients-15-04238-f003:**
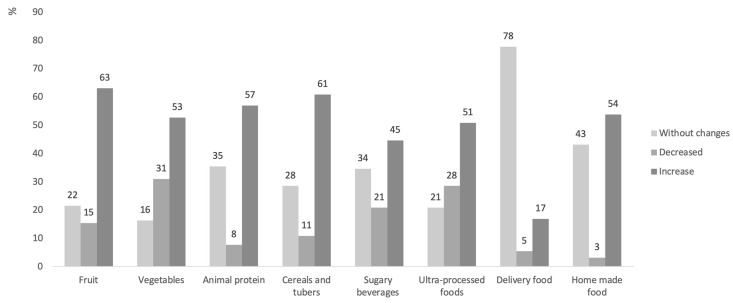
Variations in food consumption perceived by children and adolescents before and during the period of social distancing. Values expressed in percentages. Sweetened beverages include sodas, juices, and dairy products. Delivery food includes all kinds of food prepared outside the home.

**Figure 4 nutrients-15-04238-f004:**
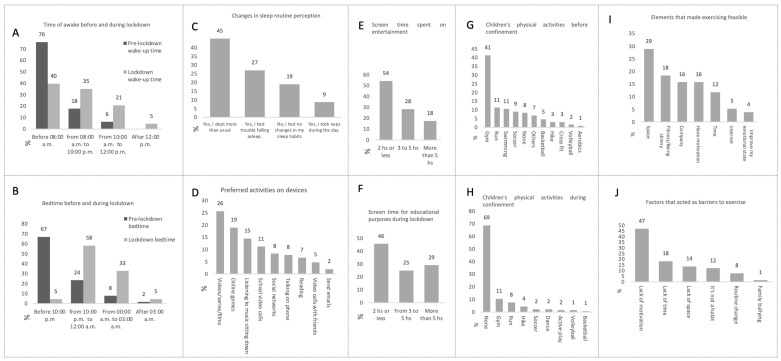
Frequencies of results on the items referring to physical activity−sedentary lifestyle and sleep of children−adolescents who are obese and overweight and their relatives surveyed at the Pediatric Obesity Clinic in the Child Well-being Unit. (**A**) Time to awaken before and during confinement. (**B**) Bedtime before and during confinement. (**C**) Changes in sleep routine perception before and during confinement. (**D**) Preferred activities on devices during confinement. (**E**) Screen time spent on entertainment during confinement. (**F**) Screen time spent on educational purposes during the lockdown. (**G**) Children’s physical activities before the lockdown. (**H**) Children’s physical activities during confinement. (**I**) Elements that made exercising feasible. (**J**) Factors that acted as barriers to exercise. Results expressed in percentages, *n* = 129.

**Figure 5 nutrients-15-04238-f005:**
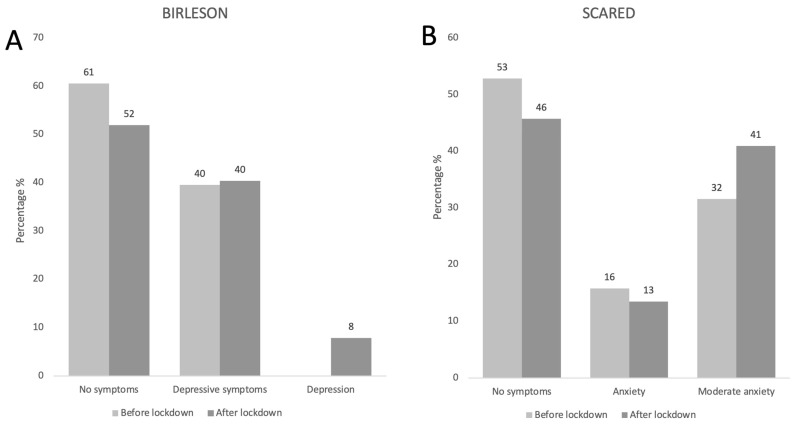
Frequency of BIRLESON (depression) and SCARED (anxiety) scale results before and during the pandemic. Results expressed in percentages, *n* = 129.

**Table 1 nutrients-15-04238-t001:** Sociodemographic characteristics of children and family members surveyed at the Pediatric Obesity Clinic in the Child Welfare Unit.

Patients	Range		Mean	SD
Age (years)	8–12	32%	13	3
	12–18	68%		
Sex (M: W)	61: 68	47%: 52%		
Overweight	25%			
Obesity	75%			
Parents/guardians				
Age (years)	18–35	34%	40	8
	36–60	64%		
	60 and over	3%		
Relationship	Grandmother	3%		
	Grandfather	1%		
	Mother	87%		
	Father	6%		
	Aunt	3%		
	Others	1%		
Educational level	Non	1%		
	Elementary school	8%		
	Junior high school	38%		
	High school or equivalent	31%		
	Degree	19%		
	Other	1%		
Marital status	Single	17%		
	Concubinage	29%		
	Married	41%		
	Divorcee	5%		
	Separated	9%		
Number of family members at home	2–4 members	57%		
	5–9 members	38%		
	over 10	5%		

Values expressed in percentages. M: men, W: women. SD: Standard Deviation.

**Table 2 nutrients-15-04238-t002:** Pre- and post-lockdown values of a subgroup of patients previously treated in the Pediatric Obesity Clinic at the Child Welfare Unit.

		Mean	SD	p
BMI	pre	26	4	**<0.001**
	post	29	5	
BMI z-score	pre	1.83	0.38	**0.01**
	post	1.96	0.39	
BMI percentile	pre	96	3	**0.01**
	post	97	3	
Waist (cm)	pre	88	14	**<0.001**
	post	97	13	
WHtR (cm)	pre	0.58	0.1	**0.03**
	post	0.60	0.1	
Fat mass (%)	pre	32	10	**<0.001**
	post	34	10	
Lean mass (%)	pre	71	8	**<0.008**
	post	66	11	
Adiposity percentage (%)	pre	23	18	**0.002**
	post	32	23	
Uric acid (mg/dL)	pre	5.6	1	0.16
	post	5.9	1	
ALT (IU/L)	pre	38	36	0.23
	post	27	20	
AST (IU/L)	pre	31	26	0.10
	post	22	8	
Total cholesterol (mg/dL)	pre	165	33	0.40
	post	156	32	
HDL cholesterol (mg/dL)	pre	39	6	0.17
	post	36	10	
LDL cholesterol (mg/dL)	pre	112	35	0.53
	post	106	32	
Triglycerides (mg/dL)	pre	156	74	0.81
	post	161	67	
Fasting glucose (mg/dL)	pre	89	7	**0.047**
	post	93	6	
Hemoglobin A1c (%)	pre	5.7	0.4	0.33
	post	5.8	0.3	
Basal insuline (mg/dL)	pre	23	16	0.29
	post	17	6	

## Data Availability

The data presented in this study are available on request from the corresponding author. The data are not publicly available due to the database is still being updated for the publication of additional works.

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
