# Peer review of "Barriers in the Management of Obesity in Mexican Children and Adolescents through the COVID-19 Lockdown—Lessons Learned and Perspectives for the Future"

_nutrients, 2023, doi:10.3390/nu15194238_

Round 1

Reviewer 1 Report

Thank you very much for this opportunity to revise the manuscript titled “Barriers in the management of obesity in Mexican children and adolescents through the COVID 19 lockdown. Lessons learned and perspectives for the future” that was submitted to Nutrients (Section - Nutrition and Obesity). 

I would like to appreciate you for performing the work on the important problem. My comments which may help you to improve the manuscript are listed below:

1.  I think it is important to provide a broader perspective by incorporating insights from other countries in the Introduction and Discussion sections. I would recommend to the authors to supplement the article with the following sources:

doi:10.3389/fpubh.2023.1168077

doi:10.1017/s0021932020000401

doi:10.3389/fped.2023.1082558

doi: 10.16469/j.css.202203008

2. I think that the sample size is too small and authors need to add a detailed argumentation.

3. It would be beneficial for the authors to provide a detailed description of the methodology used for participant selection, including the criteria for eligibility and the process of randomization. 

4. Also, the most relevant study limitations need to be appropriately described.

5. The Results section needs to be summarized to provide a more fluent section for the readers.

I am sure that the answers to these comments will improve the quality of this article.

I will be glad to review the revised manuscript.

Author Response

Thank you for your comments, please see the attachment

Reviewer 2 Report

Manuscript titled “ Barriers in the management of obesity in Mexican children and adolescents through the COVID 19 lockdown. Lessons learned and perspectives for the future.  “ is a very interesting article in the field of metabolism and COVID-19  . The overall structure is of good quality and easy to read. Methods and Results are clear and results corroborate the initial hypothesis of the authors. Figures and Tables are of sufficient quality and easy to read as well as to understand to readers. However, manuscript need some improvements, specifically in Introduction and/or Discussion. Here the points:

1. In introduction, authors should explain the risk of obesity and COVID-19 complications ( cardiovascular ) also in patients at high risk of mortality like cancer patients ( with active and not active cancer) due to increased risk factors ( cite this paper: 10.3390/cancers12113316 )

2. In discussion, authors should add non pharmacological approaches to reduce obesity and cardiovascular complications in COVID-19 patients like the use of nutraceuticals, including quercetin, resveratrol, arginine and ECGC. Especially quercetin administation, with antiviral, anti-obesity and anti-inflammatory activity should be discussed ( cite this paper: 10.3389/fphar.2022.1096853) 

Based on these improvements, this article could be accepted in this journal.

The English is of good quality and the manuscript is easy to read and understand. 

Author Response

Thank you for your comments, please see the attachment.

Round 2

Reviewer 1 Report

The authors have made good changes to the manuscript in my opinion and I appreciate how they have engaged with my reviewer comments.  I think the paper is in good shape and can, in my opinion, be published in Nutrients.

Author Response

We greatly appreciate your comments on our manuscript.

Kind regards

Eréndira Villanueva